# Current Advancements in the Molecular Mechanism of Plasma Treatment for Seed Germination and Plant Growth

**DOI:** 10.3390/ijms23094609

**Published:** 2022-04-21

**Authors:** Ryza A. Priatama, Aditya N. Pervitasari, Seungil Park, Soon Ju Park, Young Koung Lee

**Affiliations:** 1Institute of Plasma Technology, Korea Institute of Fusion Energy, 37 Dongjangsan-ro, Gunsan 54004, Korea; priatama@kfe.re.kr (R.A.P.); spark@kfe.re.kr (S.P.); 2Department of Plant Science and Technology, Chung-Ang University, Anseong 17546, Korea; aditya@cau.ac.kr; 3Division of Biological Sciences, Wonkwang University, Iksan 54538, Korea

**Keywords:** low temperature, plasma seed germination, plasma agriculture, molecular mechanism, plant growth and development

## Abstract

Low-temperature atmospheric pressure plasma has been used in various fields such as plasma medicine, agriculture, food safety and storage, and food manufacturing. In the field of plasma agriculture, plasma treatment improves seed germination, plant growth, and resistance to abiotic and biotic stresses, allows pesticide removal, and enhances biomass and yield. Currently, the complex molecular mechanisms of plasma treatment in plasma agriculture are fully unexplored, especially those related to seed germination and plant growth. Therefore, in this review, we have summarized the current progress in the application of the plasma treatment technique in plants, including plasma treatment methods, physical and chemical effects, and the molecular mechanism underlying the effects of low-temperature plasma treatment. Additionally, we have discussed the interactions between plasma and seed germination that occur through seed coat modification, reactive species, seed sterilization, heat, and UV radiation in correlation with molecular phenomena, including transcriptional and epigenetic regulation. This review aims to present the mechanisms underlying the effects of plasma treatment and to discuss the potential applications of plasma as a powerful tool, priming agent, elicitor or inducer, and disinfectant in the future.

## 1. Introduction

In recent years, low temperature plasma (LTP), also known as non-thermal plasma (NTP) or cold atmospheric plasma (CAP), has been widely applied in biology. It has broad applications in the field of biology, including seed germination, cultivation, surface sterilization, microorganism decontamination, food manufacturing and processing, wound healing, and food storage. In the field of agriculture, plasma agriculture or plasma farming involves comprehensive application of plasma to process from pre-cultivation until the product reaches the kitchen table. In plant sciences, studies on plasma treatment have been focused on exploring the possible applications, standardization of treatment, and characterization of plasma effects in terms of plant biochemistry [1]. Recently, molecular mechanisms underlying the effects of plasma on seed germination and plant growth have been explored at the cellular level, including gene expression analysis, transcriptome profiling, protein expression analysis, and epigenetics [2,3].

Seed germination is essential for the sustainability and survival of plant species. Germination is a complex process that begins with water imbibition, which triggers physiological processes, ultimately leading to the breaking of seed dormancy [4]. A rapid and uniform germination is one of the key factors for optimum crop cultivation [5]. In agriculture, numerous studies have been conducted to develop methods to promote seed germination that would eventually have a positive impact on crop growth, plant biomass, and yield. Among them, seed priming is a well-established technique for enhancing seed germination efficiency and plant growth [6,7,8]. Seed priming reduces the time required for germination and improves vigor resulting in high crop yields. The direct treatment of seeds with plasma changes the shape of the seed coat resulting in the induction of seed germination, reduced germination time, improved disease resistance, and rapid growth and development [9,10]. In addition, the production of water with altered chemical composition by plasma treatment confers antimicrobial properties, which enable uniform seed germination and induction of germination by its active components, such as reactive species [11]. This positive effect of plasma treatment resulting in seed sterilization, combined with reduced chemical and pesticide usage for pre-cultivation process, is an effective way to reduce the harmful effects of chemicals and pesticides on the environment [12,13].

Plasma treatment has been investigated in seeds of various plant species and ecotypes. In Arabidopsis (a model plant), direct plasma jet treatment promotes seed germination [14]. Moreover, irrigation of Arabidopsis with plasma-activated water (PAW, or plasma treated water, PTW) results in increased germination and an increased number of leaves and flowers [14]. Recently, research on horticultural crops and Arabidopsis is being actively conducted. It was reported that NTP enhances seed germination efficiency, improves surface sterilization of seeds, and promotes long-term growth by increasing seedling biomass, the production of antioxidants and plant hormones and the expression of genes related to defense and drought stress [15,16,17].

In radish, the germination rate was increased, and its growth was promoted when plasma was applied both indirectly—seeds irrigated with PAW—and directly [18,19]. In tomato and pepper, a combined plasma approach was used, in which seeds were exposed for 10 min (min.) to plasma and irrigated with PAW, and the germination rate and plant growth were found to be increased compared with that in samples treated with water [20]. Recently, in ginseng, plasma treatment caused an increase in germination rate and sterilization, and also inhibited the growth of microorganisms exhibiting antifungal activity, especially showing an inhibitory effect on root rot [21]. Therefore, direct and indirect plasma treatments of seeds are considered to positively affect the plant development, including seed germination, growth, and disease prevention. However, optimal conditions in terms of intensity and treatment duration of plasma depending on the variety and ecotype of the crop are crucial factors that would determine the success of this technique for future applications [22,23].

Although studies on plasma have been rapidly reporting on its new potential applications for seed germination and plant growth, the molecular mechanisms underlying the effects of plasma treatment on plants are relatively unexplored. Therefore, in this review, we have summarized the recent progress in plasma treatment for seed germination and plant growth and in understanding the molecular mechanisms underlying the effects of the treatment. First, we have discussed plasma and plasma treatment applied in biology, especially for improvement of seed germination and plant growth. Additionally, we have described how plasma treatment was developed and utilized, including the technical advancements and challenges in recent years. Second, we have discussed the current understanding of physical and biochemical mechanisms underlying the promotion of seed germination and plant growth. Third, we have summarized current studies based on the molecular aspects related to the effects of plasma treatment on seeds and plants. Finally, the future prospects including future research, possibilities, and challenges are discussed.

## 2. Plasma Treatment Method

Plasma is a partially or fully ionized gas that can be ignited at low atmospheric conditions and consists of charged species (electrons and negative and positive ions), neutral species (atomic and/or molecular radicals and non-radicals), electric fields, and photons. One of the earliest applications of plasma for treatment of seeds was studied in early 1960s, when the effects of glow discharge on cotton, wheat, alfalfa, red clover, sweet clover, beans, and several varieties of grass seeds were investigated. It was shown that the plasma treatment influences seed germination, moisture adsorption, and apparently reduces hard-seed content in legumes [24,25,26,27]. Since then, studies on plasma treatment of seeds have been expanded by using various kinds of plasma devices, which allow detailed studies on physical, chemical, and biological mechanisms of plasma that can be triggered by analysis of plasma components [28].

In the past decades, LTP at atmospheric pressure has opened up a new research field in biology and medicine [29,30]. Plasma treatment of seeds has been divided into two methods—direct and indirect—based on the contact of the plasma with the samples. For plasma treatment, plasma sources, such as dielectric barrier discharge (DBD) [31,32,33,34,35,36,37], radio frequency (RF) plasma [38,39,40,41,42,43,44], and atmospheric pressure plasma jet (APPJ) [15,45,46], have been used (Figure 1). The treatment is performed by controlling operating parameters, such as electrode structure, power source (voltage, frequency, and waveform), discharge gas (air, Ar, He, etc.), and other conditions (gas flow, gas pressure, gas temperature, etc.).

In direct plasma treatment, the plasma devices usually consist of a place to plant seeds in the container module, where they are exposed to the plasma generated by the generator and electrode. Because the surface area of Surface DBD (SDBD) is relatively wider compared with that of other plasma devices, it has been more commonly used than other plasma devices for biological applications. For example, SDBD plasma device has been widely used in the germination and growth of Arabidopsis, barley, bell pepper, maize, pea, quinoa, and wheat [33,47,48,49,50,51]. The exposed seeds in the discharge area are directly affected by charged particles, reactive species (such as OH radicals, singlet oxygen, ozone, and hydrogen peroxide), electric fields, and photons (visible/ultraviolet (UV) radiation). A combination of these components is believed to be the main factor that promotes seed germination and growth. During the exposure, the seed surface interacts with the short lived and long-lived radicals, which appear due to secondary reactions.

In indirect treatment, the sample is not exposed to the plasma itself. The plasma does not directly affect the samples, but a gas-phase active species generated by the plasma and PAW affect the sample. Plasma-treated water causes changes in the physicochemical properties and PAW participates in the signaling pathway and eventually promotes seed germination, root and vegetative growth, and plant reproduction [52,53].

PAW, especially under atmospheric conditions, is known to change the electrical conductivity, pH, concentration of nitrite (NO_2_^−^), nitrate (NO_3_^−^), ozone (O_3_), and hydrogen peroxide (H_2_O_2_) [54]. These changes in physicochemical properties and components mainly contribute to the benefits of plasma treatment for seed germination and plant growth. Most research on seed treatment has been focused on the inhibition of microbial growth on the seed surface. The application of PAW for inhibition of microorganisms is usually linked to the increasing acidity with the duration of plasma treatment [55,56,57]. Moreover, the reactive oxygen and nitrogen species (RONS) content is increased, which also inhibits microbial growth that later affects seed germination and RONS functions in seed priming [10].

In addition to seed priming, the high nitrate and nitrite content in PAW is believed to be the major factor contributing to the improvement of plant growth under PAW irrigation because it can act as a substitute for nitrogen source. Hence, the concept of “plasma-fertilizer” was established. This concept is considered important because nitrogen is the backbone of all metabolic processes that directly affect plant growth. In addition, “plasma fertilizer” is one of the eco-friendly alternatives of nitrogen source that reduces the disadvantages associated with the use of chemical fertilizers. Therefore, it is not surprising that this is the major topic for future studies on plasma treatment for plant cultivation [58,59,60,61].

## 3. Chemical and Physical Effects of Plasma Treatment

Direct plasma treatment of the seed surface is the most common treatment performed in seed germination studies. When the plasma contacts the seed surface, it changes the seed coat, resulting in reduced germination time and rapid growth and development [14,62]. The positive effects of plasma treatment of seeds have been observed in several species of horticultural crops [15]. Plasma studies in horticultural crops show that atmospheric plasma increases the germination rate and growth rate in radish, pepper, and tomato in correlation with the duration of treatment [20]. In addition to the germination of tomato seeds, use of helium plasma for seed priming increases its overall growth compared with that in control [15]. It is well known that plasma treatment of seeds is mainly affected by RONS. The production of RONS from plasma is also regarded as a major factor in enhancing seed germination, seedling growth, and plant defense. In addition, seed surface or seed coat changes are categorized as mechanical changes. Moreover, factors such as seed sterilization, heat, UV irradiation, ionization, and electromagnetic fields generated by plasma devices are also believed to play roles in seed germination induced by plasma treatment. Therefore, in the following section, we have summarized the regulation of the plasma treatment of plants with regard to these aspects (Table 1, Figure 2).

### 3.1. Chemical Effects of Plasma Treatment

Several terms have been used in literature for reactive species to describe the oxygen radicals and non-radicals. Reactive species are also grouped into reactive oxygen species (ROS: O_2_^−^, H_2_O_2_, O_3_, etc.) and reactive nitrogen species (RNS: ^•^NO, ^•^NO_2_, ONOO^−^, etc.). Some common reactive species include ozone (O_3_), hydrogen peroxide (H_2_O_2_), superoxide anion, peroxyl, nitric oxide, nitrogen dioxide, and peroxynitrite [63,64,65]. Reactive species generated from plasma treatment are believed to be the major factors affecting seed germination and plant growth. However, mechanisms underlying the effects of ROS and RNS on seed germination and development are not fully understood. There are few hypotheses about how the external ROS affect seed germination. One hypothesis is that external ROS are detected and perceived by the cells in seeds that induce signal transduction from the outer layer of the seeds. The other hypothesis is that during imbibition, water is the key factor in the absorption of ROS into the cell layers of the seeds. Thus, it increases the respiration of seeds and triggers a chain reaction of sugar oxidation to release metabolic energy in the form of ATP [47]. Therefore, the involvement of ROS in the respiration pathway is considered a primary and secondary trigger in seeds that causes transition from dormancy to metabolic activity.

A previous study showed the presence of external ROS in certain amounts in water during imbibition and also showed that wet seeds may trigger faster signaling in the intercellular process; however, the effect of ROS on dry seeds is hardly understood. There is a possibility that the effect of ROS on dry seeds is minimized or delayed. During plasma treatment, ROS penetrates the seeds, but no specific mechanism occurs until the start of imbibition [66]. However, how ROS are stored during the period before imbibition remains unknown. In addition, this theory does not fully explain how long-term storage after plasma treatment still a positive effect on the germination and growth of seeds compared with those of untreated seeds, especially when ROS are known to be mostly short-lived [67]. Biochemical changes in plasma-treated seeds apparently continue to occur even long after the seeds are treated [20,68]. These changes are related to gene expression, the oxidative process, protein concentration and hormones. It is believed that plasma treatment increases the pore size of the seed coat, which increases water imbibition and ROS absorption, leading to genetic regulation of seeds [69]. Other possible ways by which seeds and plants might absorb RONS generated from plasma treatment could be through the bypassing of protein channels, named aquaporins, which are mainly used for water transport [70]. It is important to note that, in addition to transport via aquaporins, ozone is absorbed through stomata in the seedlings and mature plants. Therefore, it has been observed that the accumulation of ROS in leaves through stomata and micropores is the main pathway by which ROS can travel further to other plant tissues. It was also shown that excessive absorption of ROS by plants could result in chloroplast damage [71].

A relatively small amount of H_2_O_2_ (0.12 ppm) in PAW, generated from a plasma device, increased the seed germination rate in tomato and pepper seeds [20]. In Arabidopsis, PAW containing 17–25.5 mg/L H_2_O_2_ had a positive effect on germination and seedling growth [14]. Among RNS, low nitrate concentration (100 ppm or less) enhances seed germination and seedling growth in plants, but the growth tends to be inhibited above 100 ppm [20,72,73,74]. However, plants apparently have their own nitrate and ROS sensitivity and show a dosage-dependent growth pattern.

One of the prominent members of reactive species, which is abundantly detected in plasma treatment, is hydrogen peroxide (H_2_O_2_). A previous study on the involvement of H_2_O_2_ in seed germination and seedling growth revealed its role in the absorption mechanism, signaling pathway, regulation of gene expression, protein modification, and other related factors [75]. The absorption of H_2_O_2_ into the cells occurs mainly through diffusion and is dependent on the anion channels; inside the cell, it breaks down into singlet oxygen and hydroxyl, thus, allowing easy transfer between cells [70,76]. During germination, H_2_O_2_ mediates the regulation of abscisic acid (ABA) catabolism and gibberellic acid (GA) biosynthesis [77]. In addition, upon exposure of seeds to external H_2_O_2_, the endogenous H_2_O_2_ levels also increase and induce several oxidative pathways, such as carbonylation and lipid peroxidation [78]. Moreover, the presence of H_2_O_2_ in cells is regarded as a priming factor that involves complex changes in the proteome, transcriptome, and hormone levels [78].

Ozone, as one of the major ROS, was shown to be responsible for improved seed germination and induction of protein expression in seeds [79,80]. Ozone generated during the presence of UV radiation can eventually generate superoxide and hydroxyl radicals in the seed coat, which could be one of the main reasons for how the combination of external physical damage of the seed coat combined with chemical stimulus of the accumulated radicals works synergistically to increase germination. However, it is important to note that different device configurations produce different concentrations of ozone. Moreover, the purpose of treatment also determines the ozone concentration required. Postharvest treatment with 0.3 ppm ozone in combination with cold storage could inhibit the decay process and reduced severe infection in peach and table grapes [81,82]. In strawberry, plasma treatment using different sources of gas has been investigated; it was observed that plasma treatment for 5 min could produce 600–2800 ppm ozone that had a positive effect on microbial disinfection and strawberry freshness [83]. Similarly, various ozone concentrations were investigated in the plasma treatment of seeds. In Arabidopsis seeds, the effects of treatment with 200 ppm ozone, generated from a plasma device, for 10 min on seed coat modification were studied [84]. A low concentration of ozone (~1–5 ppm) is effective in plant growth by killing larvae in the soil and in fresh cut green leaf lettuce [85]. Interestingly, a similar concentration of ozone (~1–4 ppm) generated from surface discharge successfully reduced the number of nematodes and induced plant growth [86].

The biochemical mechanism underlying the effects of plasma treatment on seeds is very closely related to the metabolism of antioxidant enzymes. The seed coat may contain proteins or enzymes, such as NADPH oxidase, superoxide dismutase (SOD), and peroxidase (POD), which can convert the substrate into signaling molecules such as H_2_O_2_. Plasma treatment of wheat seedlings resulted in increased isoenzyme activities, such as POD and phenylalanine ammonia lyase (PAL), which are crucial for the production of polyphenols, which also participate in plant defense [87,88].

Another area of focus in plasma treatment is the potential application of RNS produced from plasma–liquid interaction as a liquid fertilizer for plant growth. Various approaches have been explored to obtain the most suitable device and treatment method for the production of high amounts of RNS in the solution. For example, a large volume of glow discharge has been tested as a liquid fertilizer in radish, tomatoes, and marigolds [89]; bubble discharge has been investigated in the cultivation of spinach, radish, *Brassica rapa*, and strawberry [90,91,92]; a plasma jet has been used for plasma-assisted nitrogen fixation for corn [93]. Collectively, these reports demonstrate the potential of plasma fertilizers as an alternative and a more eco-friendly approach for nitrogen source for plant cultivation. However, one of the challenges in plasma-assisted nitrogen fixation is the low pH or increased acidity of the solution treated with plasma, which damages the seed and plant exposed to such solutions. Plant growth is limited under acidic environment [94]. Therefore, studies on controlling the balance and on methods to overcome the acidity of plasma-activated solutions are being considered a priority in the plasma field. Lamichhane et al. recently demonstrated an innovative approach to control the acidity of plasma-treated water using a combination of chemical additives including Mg, Al, or Zn, which could neutralize via the reduction in pH [93]. Moreover, the presence of these additives increases the rate of reduction of nitrogen to ammonia, which results in the improvement of germination rate and seedling growth [93].

Inactivation of microbes by plasma treatment has been used as the fundamental technology in medicine and food processing [95,96,97]. Microbial inactivation, resulting from plasma treatment of seeds, also plays important roles in germination. The seed surface is usually exposed to the environment, which contains many types of particles, contaminants, and microbes that could have negative effects on seed germination. For example, in grain crops, such as rice, wheat, oat, and barley, the growth of *Fusarium* on the seed surface affects germination [98]. It is known that fungal infections on seeds often damage their viability and potentially reduce the yield. Moreover, fungal pathogen on seeds can also lead to a seed-borne infection, which can cause abundant yield loss. The plasma treatment of seeds has been shown to have positive effects on seed sterilization, including removal of fungal spore on the seeds. In rice, direct treatment using micro corona discharge of SDBD can inactivate the microorganisms on the husk, which leads to higher germination compared with that in untreated seeds [99]. This finding emphasizes the incorporation of Ar and air in plasma that would result in the production of ROS and RNS, which decontaminate and inactivate fungi on the seed surface. Another study using indirect treatment with arch discharge plasma showed successful inactivation of *Fusarium fujikuroi* (a fungal pathogen) in a submerged rice seed suspension; however, fungal spores are more effectively inactivated using ultrasonic waves as a source of ozone and shock-wave [100]. Collectively, these two examples show that both direct and indirect treatments are effective in inhibiting the fungus through the production of ROS and RNS. Moreover, positive results for microbial inactivation or sterilization of seeds were also confirmed in many different seeds, such as wheat, barley, oat, lentils, maize, chickpea, sunflower, and scots pine [12,101,102,103,104].

The mechanism underlying the effects of plasma treatment in the inactivation of microorganisms has been well-investigated using different device types, exposure times, power, gas sources, and other factors. Compared with conventional seed sterilization using active chemicals, the mechanism of plasma disinfection is often considered complex due to variations in devices and plasma components [105]. However, the general agreement for the use of plasma treatment for disinfection is due to the production of reactive species, exposure to which triggers a complex sequence of events in the microbe, resulting in the antimicrobial activity [106]. For example, exposure to ROS directly triggers molecular damage in cells, including DNA breakage, lipid peroxidation, and carbonylation of proteins [107,108]. Moreover, combined action of RNS and ROS is important for antibacterial activity. The effect of plasma with high content of ROS and RNS is more than that of plasma-treated water on the antimicrobial activity [109]. UV exposure also plays an important role, especially in the inactivation of bacteria using plasma treatment. It is known that mechanism of action of UV exposure may be related to several specific mechanisms, including direct destruction of genetic material, breaking of chemical bonds in organic compounds, and the phenomenon of plasma etching (UV-induced etching) [110,111,112].

Other applications of plasma treatment include the inactivation of viruses. In the field of plasma biomedicine, the inactivation of coronavirus using plasma devices has recently garnered a lot of attention and proved to be effective in treating viral infection and associated diseases [113,114,115]. In plants, potential applications of plasma treatment of virus-related diseases have been reported. Potato virus Y (PVY) homogenized in water was successfully inactivated by plasma treatment for 1 min [116]. A sample of pepper mild mottle virus (PMMoV) in water was also successfully inactivated using 99% argon and 1% oxygen plasma jet in 5 and 3 min [117]. As for the inactivation of bacteria, the mechanism of virus inactivation is primarily through the production of reactive species (ROS and RNS) with various physical effects, and the treatment could damage the virus particle and degrade viral DNA/RNA [118,119].

### 3.2. Physical Effects of Plasma Treatment

Plasma treatment of seeds, especially direct treatment, is considered to have a similar principle to plasma etching on the seed surface [120]. It is shown that seed coat modifications are presumably highly dependent on the type of plasma devices, power, and duration of the treatment. For example, treatment of Arabidopsis, wheat, radish, and oat seeds showed no distinct surface modifications based on seed coat morphology [121,122,123,124]. However, other studies revealed seed coat degradation by plasma treatment, which was observed using scanning electron microscope (SEM), in different plant seeds such as Arabidopsis, cotton, wheat, mimosa (*Mimosa caesalpiniaefolia)*, erythrina (*Erythrina velutina)*, pea, and onion [34,35,125,126,127,128,129]. Therefore, the effect of plasma treatment on seed germination may be due to the mechanical factors of seed coat modification, especially when treated with appropriate plasma device and configuration.

The seed coat protects the seed from the external environment and regulates the water absorption. Imbibition must occur in the correct ratio; otherwise, seeds may be damaged if imbibition is too slow or too fast [69]. Plasma treatment affects seed germination differently in different seeds of various species and different plant families, even for different variety and ecotype, and generally, a different setup is needed for optimum treatment condition. The seed coat consists of cuticle, epidermis, hypodermis, and parenchyma cells. The degradation of cuticle layers will allow water to be absorbed further into inner layers. Plasma treatment helps in the removal of lipid layers on the cuticle and epidermis, which accelerate the germination [130]. In Arabidopsis seeds, modifications in the composition of lipid compounds in the seed coat was detected after plasma treatment [37]. Interestingly, Arabidopsis seed coat mutants, *gl2* and *gpat5*, and Col-0 were examined under plasma treatment; the germination rate was increased in plasma-treated seeds, even under osmotic and saline stress conditions, whereas the germination ratio in *gpat5*, with defective cuticle layers on the seeds, was not rescued by plasma treatment due to less permeability and more sensitivity to plasma processing. POD activity in testa and endosperm tissues was detected, where the major POD function occurred in ruptured seeds. This shows that in plasma-treated seeds, the structure and composition of lipid compounds are changed before germination and metabolism changes after germination [37].

Seed surface modifications are usually investigated using light microscope or SEM. In addition, biochemical analysis is also performed to confirm the differences in seed coat content, such as lignin, cutin, polysaccharides, and other ROS-related proteins. Indirect treatment using plasma-treated water was also performed to determine the seed wettability, which showed possibilities of combined mechanism involving seed perforation and lower water tension that increases the surface area and ability of imbibition. Seed coat modifications are closely related to water permeability and water affinity on seed surface, also known seed wettability [131]. The increase in seed wettability improves the permeability of seeds to water, and thus, the imbibition process is accelerated. The wettability has been known to be majorly affected by the morphology and chemistry of seeds. Morphologically, plasma treatment causes surface etching or surface erosion that increases the roughness of the seed surface, resulting in the increase in seed volume ratio, and thus, wettability is increased. Chemically, the interaction of plasma and seed coat components affects the organic polymers in seeds, for example, it results in degradation of cutin and wax layers, thus reducing the hydrophobic activity of the seed coat and increasing water permeability [131,132,133].

The heat effect of plasma treatment was examined by comparing the plasma treatment with the heat plate treatment, which shows different effects on germination [121]. It was shown that heat is not the main factor contributing to the increase in germination of plasma-treated seeds. Although plasma treatment is regarded as “non thermal plasma or cold plasma”, the slight increase in temperature in plasma-treated seeds cannot be ignored as it may still affect the germination ratio when combined with other factors. Heat shock proteins (HSPs) are one of the important regulators of plant metabolism. They sense temperature changes, perceive signaling, and respond to protect other proteins from stress-induced damage [134,135,136]. Iranbakhsh et al. confirmed that heat resulted in the induction of HSP expression in plasma-treated wheat seeds [36]. However, it is difficult to distinguish whether heat increased the scarification of seeds, which led to the induction of germination and indirectly increased the expression of HSPs, or whether the heat generated by plasma treatment directly induced HSPs [18]. Moreover, in a study on plasma-treated sunflower seeds, no change in the expression of HSPs was evident in proteome profiling and an ambiguous result that the treatment may not induce significant accumulation of pathogenesis-related (PR) proteins or HSPs was obtained [42]. Remarkably, a recent study on the expression of HSP genes in maize grain treated with diffuse coplanar surface barrier discharge (DCSBD) showed induction of several HSP genes, including *HSP101*, *HSP70*, and *HSF17*, that affected grain vitality and seedling growth [137].

Ultraviolet light is regarded as one of the main factors in plasma sterilization, especially in the inactivation of surface microbes [96,138]. Single spectrum UV lights, such as UV-A, UV-B, and UV-C, have been shown to increase seed germination but act differently during seedling and plant growth. Exposure of *Amberboa ramose* seeds to UV-A resulted in increased germination and growth during plant development [139]. UV-B was shown to increase the germination rate of safflower (*Carthamus tinctorious*), radish, cabbage, kale, and agave seeds [140]. However, prolonged exposure to UV-B inhibits the growth of seedlings [140,141]. UV-C treatment in maize and sugar beet was also shown to increase germination rate and seedling growth [142]. During plasma treatment, the possibility that UV light directly affects seed germination is considered very low due to its short exposure duration and low intensity of irradiation from the plasma sources. In some studies, it was proposed that UV irradiation due to plasma treatment may have indirect effects combined with RONS interaction and not by improvement in seed wettability [125,143]. In addition, the nature of UV exposure could induce DNA damage, which could also affect germination and seedling growth. Short-term exposure of seeds and seedlings to UV could induce their growth by regulation of stress response, especially when the irradiation is perceived by the photoreceptor, leading to increased cell metabolism, including cell differentiation, division and elongation [144].

## 4. Molecular Mechanism of Plasma Treatment

### 4.1. Regulation of Reactive Species

The molecular mechanism underlying the effects of reactive species on seed germination and plant growth has been well characterized. It is known that RONS directly and indirectly regulate gene expression during seed germination [174,175,176,177]. In the plant–plasma field, reactive species have become the major factors in plasma-induced developmental and stress responses, including sterilization, germination enhancement, seedling and plant growth, stress, and plant defense responses. The presence of RONS affects the regulation of several molecular pathways in seeds and seedlings, such as oxidative pathway, reductive pathway, ABA catabolism, and GA biosynthesis [177,178]. Molecular studies on plasma treatment began with the investigation of enzymes involved in the oxidative and reductive pathways, such as POD, polyphenol oxidase (PPO), ascorbate peroxidase (APX), catalase (CAT) and SOD. In addition, hormone content is often quantified to identify the role of reactive species in regulating hormone pathways. The effect of plasma treatment on enzymes related to redox homeostasis has been investigated in various plant species, such as Arabidopsis, tomato, rice, wheat, radish, barley and sunflower [14,15,143,170,179]. Generally, the results show the induction of redox enzymes, which then regulate downstream gene expression as well as signaling and transcriptional regulation [15,180].

Reports on the molecular regulation of reactive species in plasma-treated plants are relatively scarce. Adhikari et al. reported plasma-induced seed priming increased germination of tomato seeds after direct treatment using a plasma jet. Further examination of the seedlings showed improved plant growth and changes in the biochemical markers of stress-related genes and enzymes. The regulation of antioxidant genes (*POD*, *PPO*, *CAT*, *glutathione transferase*, and *SOD*), pathogen resistance genes (chitinase, β-1,3 glucanase and basic glucanase), and phytohormone synthesis genes (alternative oxidase, *AOX*; 12-oxo-phytodienoic acid reductases, *OPR*; and phenylalanine ammonia lyase, *PAL*) was modulated by plasma treatment. In addition, the expression of respiratory burst oxidase (RBOH) and mitogen activated protein kinase (MAPK) was focused on because their function is crucial in the signaling cascade of multiple pathways [15]. Similar findings of the upregulation of antioxidant genes in maize were also reported in the enhancement of seed germination and growth in maize by plasma treatment using low-pressure dielectric barrier discharge (LPDBD) [181]. The LPDBD plasma generated from argon/air (Ar/air) increases germination rate and growth by elevating ROS-related enzymes in seeds and plants. This increase is then maintained by the upregulation of expression of Z*mCAT* and *ZmSOD* and the activity of these genes ultimately induces seed germination and plant development [181].

The focus of molecular studies conducted on plasma treatment has not only been on seed germination but also on plant defense. It is expected that the molecular mechanism underlying plasma-dependent defense response involves an RONS-related pathway. Tomato seedlings irrigated using PAW induce the expression of MAPK gene. It is predicted that the MAPK signaling cascade activates expression of several genes regulating plant defense response, such as *β-1,3-glucanase* and *chitinase*
*3 acid*. Furthermore, genes involved in the biosynthesis of plant defense-related hormones, salicylic acid (SA) and jasmonic acid (JA), were also induced; these included 2-oxophytodienoic acid reductase (*OPR1*), allene oxidase synthase (*AOS*), and phenylalanine ammonia lyase (*PAL*) [180]. Therefore, it is concluded that PAW can induce an immune response, but further detailed studies are needed to confirm its role in the RONS signaling pathway.

RNS produced upon plasma treatment have also gained significant attention, especially due to their potential function as fertilizers as they are a good nitrogen source [28,89,182]. To date, many studies have reported that high amount of nitrate in plasma-treated liquid increases the plant yield and biomass when used for regular irrigation during plant growth. However, few studies have focused on the roles of RNS at the molecular levels. Interestingly, Lee et al. treated tobacco plants with PAW using SDBD and found phenotypic changes in both vegetative and root tissues; they showed that at the cellular level the duration of plasma treatment is correlated with the formation of root hair and palisade cell size [73]. Increased lateral roots and root hair under PAW treatment are related to the overexpression of *Nicotiana tabacum* Cobra-Like (*NT-COBL*), NT xyloglucan endotransglucosylase/hydrolases 9 (*NT-XTH9*), and NT Xyloglucan endotransglucosylase/hydrolases 15 (*NT_XTH15*), which are involved in root hair development and cell size regulation. In addition, PAW regulates root hair density in Arabidopsis by shortening root hair length but dramatically increasing the number of root hair in the same root space [72]. Under optimal conditions, PAW treatment for 5 min increased the root length in Arabidopsis by 87% compared with that in the control. A dramatic increase in root length and root hair number was caused by positive modulation of expression of Arabidopsis COBRA-LIKE 9 (*COBL9*), AUXIN 1 (*AUX1*), LIKE-AUXIN1 3 (*LAX3*), *XTH9*, and *XTH17* [72]. PAW-mediated molecular changes in root architecture occurred due to the modulation of specific genes, especially those related to root development, plant phytohormones, and cell wall modification. A long-term effect of DBD plasma-treated soybean seeds is the enhancement of nitrogen fixation nodules in soybean roots, which increased by 63% and improved nitrogen fixation efficiency in roots. At the molecular level, the expression of soybean Expansin 1 (*G**mEXP1*), which is found in plant cell walls and promotes cell wall loosening, was upregulated in plasma-treated samples [163]. Based on these findings, the molecular regulation by plasma via the induction of cell expansion is one of the fundamental regulations, which led to the increase in cell size, plant organs, and overall plant development.

### 4.2. Effect of Plasma on Stress Response

Plasma treatment could affect gene expression during seed germination and growth, which could lead to changes in protein expression. The DBD plasma treatment promotes ABA production in wheat seedlings under drought conditions, and plasma treatment regulates drought resistance related genes—*LEA1*, sucrose nonfermenting 1-related protein kinase 2 (*SnRK2*) and pyrroline-5-carboxylate synthetase (*P5CS)* gene. Plasma treatment stimulates the expression of regulatory genes, sucrose nonfermenting 1-related protein kinase 2 (*SnRK2*) and pyrroline-5-carboxylate synthetase (*P5CS*), while it decreases *LEA1* transcript levels under drought conditions [183]. In addition, regulation by transcription factors via turning on and off of gene expression by binding to specific sequences was recently observed in plants. Gene expression of transcription factor, heat shock factor A4A9 (*HSFA4A*), in wheat was first induced in roots after 3 h of plasma exposure, which resulted in resistance to salt stress [36]. Heat shock factors (HSFs) are well-known transcription factors in plants that act in response to heat stress and regulate plant development and stress response. They can bind to the heat shock elements (HSEs) and regulate the expression of HSPs and are involved in MPK3/MPK6 signaling. These results suggest that plasma recognizing and signaling pathways can create some degree of stress environment for the seed and cause changes in gene expression at the transcriptional level, thereby, enabling seed priming with plasma to enhance seed germination, growth, and resistance to biotic and abiotic stresses.

### 4.3. Hormone and Metabolite Regulation

Plant hormones, also known as phytohormones, regulate plant growth and development as endogenous organic substances by providing developmental cues while also being affected by environmental cues for seed germination. Plant hormones are regarded as the main regulators of seed germination and seedling growth, and include ABA, GA, auxin, SA, and cytokinin. The presence of ABA in seeds maintains their dormancy, whereas the increased GA content in seeds promotes the initiation of seed germination. Previous studies have shown that the plasma treatment of seeds may directly and indirectly affect the hormone content in seeds [35]. Mildaziene et al. reported that RF plasma treatment disturbs the phytohormone balance and protein content, which may lead to the promotion of seed germination [42].

The effect of plasma on the regulation of plant hormone-related gene expression has attracted the interest of many research groups in view of the endogenous nature of plant hormones and their roles in the regulation of plant growth, development, seed reproductivity, pathogen defense, and abiotic and biotic stresses. A major plant hormone, ABA, is called a stress hormone as it alleviates stress stimuli and regulates plant growth, development, and various stress responses including drought, cold, and radiation [184]. After seeds are exposed to plasma, ABA fluctuation in freshly harvested seeds impacts the germination and results in decreased ABA and increased GA content in radish [185]; soybean seed quality was regulated by seed-borne pathogen fungi and fungi damage could be rescued by plasma treatment, which affected ABA and IAA content in seeds [186]. Additionally, plasma treatment enhanced cold tolerance in tomato via ABA signaling and H_2_O_2_ involvement mediated by respiratory burst oxidase homolog 1 (*RBOH1*). H_2_O_2_ mediated by *RBOH1* played a role in downstream mechanism caused by plasma treatment, which involves increase in ABA content by upregulation of 9-cis-epoxycarotenoid dioxygenase 1 (*NCED1)* that encodes 9-cis-epoxycarotenoid dioxygenase enzyme involved in the ABA biosynthesis pathway. This plasma induced H_2_O_2_ and ABA signaling cascade increased the expression of regulatory genes, CBF expression 1 (*ICE1*) and C-repeat/dehydration responsive element binding factor1 (*CBF1*) [165]. Recently, plasma treatment was shown to increase ABA concentration in Arabidopsis seedlings by modulating the expression of ABA and ROS-related genes, and it also mediates intracellular ROS production by inducing expression of *RBOH1* genes, which are key for ROS production. Accumulated ABA also increased Ca^2+^ concentration, which reduced stomatal opening [187]. In plasma treatment, ROS and Ca^2+^ might function as secondary signaling molecules to enhance growth of seedlings. In addition, after plasma treatment, heat-stressed rice seeds recovered from delayed germination by epigenetic regulation of ABA biosynthesis, but not of GA biosynthesis, and signaling [187]. Plasma-treated rice seeds subjected to heat stress show significant hypermethylation of *OsNCED5* promoter and hypomethylation of *OsAmy1C* and *OsAmy3E* promoters, which are involved in the biosynthesis and catabolism of ABA, respectively. This pattern of methylation is consistent with their gene expression pattern [162].

The modulation of plant hormones, including SA and JA, which are commonly known as defense hormones, has been investigated in several plasma treatment studies. Under abiotic and biotic stress, plants produce secondary metabolites, endogenous plant hormones, and RONS. Induced endogenous substances increase the expression of plant defense-related genes, PR genes, and hormone (SA or JA) biosynthesis genes. Tomato seedlings grown from seeds treated with plasma jet showed a fluctuation in SA and JA content. Both JA and SA were increased at 10 min after treatment, whereas at 5 min after treatment, a significant reduction in SA, but no change in JA content, was observed [15]. In maize, direct treatment of seeds using a plasma jet showed both negative and positive effects on SA and JA levels in the seedlings depending on the treatment duration [188]. Interestingly, another study using PAW irrigation in tomato seedlings showed increased SA content in shoots and roots, whereas JA was modulated depending on the duration of PAW treatment. This induction is predicted due to the regulation of MAPK signaling cascade caused by the influence of H_2_O_2_ and NOx content, which are exogenously supplied by PAW [180]. Plant MAPKs are more similar to the mammalian extracellular signal-regulated kinase (ERK) subfamily of MAPKs and their cascade plays a vital role in plant defense signaling under biotic and abiotic stress. MAPKs include the signaling molecules—MAPK kinase kinase (MAPKKKs) activate MAPK kinase (MAPKKs), which eventually activate MAPKs [189,190]. H_2_O_2_ transduces signal via calcium as the secondary messenger and the MAPK cascade. RONS (NO and H_2_O_2_) produced by PAW irrigation induce plant hormones, SA and JA, and PR proteins via the MAPK signaling. At transcriptional level, upregulation of MAPK related gene expression was observed under PAW irrigation [68,180].

Taken together, plasma treatment may induce a fluctuation in the SA and JA levels followed by a plasma-induced RONS-mediated signaling cascade in plants. Because plasma treatment generally induces beneficial metabolic products related to stress response pathways, some studies focused on the investigation of stress response markers that correlated with the increase in the polyphenol content of plasma-treated plants. For example, ginseng seedlings grown from plasma-treated seeds had increased polyphenol content (approximately 56%) compared with that in the control [169]. Although the exact mechanism has not been well characterized, the expression of defense marker genes was induced in the plasma-treated sample [169]. Moreover, effect of plasma treatment on the expression of phenylpropanoid genes has been reported in basil (*Ocimum basilicum*). Direct treatment of basil seeds using DBD plasma increases the expression of three genes in the phenylpropanoid biosynthesis pathway—cinnamate 4-hydroxylase (*C4H*), 4-coumarate coA ligase (*4CL*), and chavicol O-methyl transferase (*CVOMT*) [191]. Additionally, plasma-treated cannabis seeds for priming exhibit upregulated gene expression of secondary metabolites. The expression of genes regulating olivetol-related genes, including olivetol synthase and olivetol acid cyclase, was reported to be increased 42-fold. Cannabinoid acid synthase gene was increased approximately 26-fold after plasma treatment. In addition, plasma priming upregulated the expression of *WRKY1,* which is a transcription factor involved in biotic and abiotic stress response, such as mold, low temperature, salt, and drought resistance [146]. In fenugreek, plasma treatment resulted in an increase in chlorophyll and carotenoid content and induced the expression of genes related to the biosynthesis of diosgenin [172]. To date, various studies have shown an increase in the content of secondary metabolites by plasma treatment, such as in ginseng, basil, coneflower, and carrot [145,153,169,192]. However, most studies did not provide the molecular mechanism underlying the secondary metabolite induction. According to our current understanding, plasma may play a role in the induction of metabolites that modulate stress-related pathways as an elicitor or an inducer. Therefore, a detailed investigation of its potential applications and further characterization of secondary metabolite production are necessary.

### 4.4. Transcriptome and Proteome Profiling

A common method in molecular biology for measuring gene expression at the transcriptional level is qRT-PCR, which allows the analysis of individual target genes of our interest [193]. In the past decade, the integration of systems biology has provided opportunities to study a complex mechanism by unveiling the expression pattern in various biological systems. The studies have utilized the combination of biology, technology, and computation to reveal mechanisms behind various biological phenomena. In the field of plasma application in seed germination and plant growth, only a few studies have utilized systems biology tools, such as transcriptomic, proteomic, and metabolomic analyses [170,171,194,195]. These approaches can provide different perspectives on the effects of plasma application at the molecular level.

In Arabidopsis, gene expression profiling under plasma treatment has been investigated using microarray analysis. Watanabe et al. investigated the effect of different gas feeders, including different pressures of argon and oxygen, on Arabidopsis and radish seeds, followed by microarray analysis using RNA extracted from seeds. Interestingly, active reactive species from plasma irradiation affect the plant cell size. Based on microarray data, they suggested that one of the factors was related to oxidative stress and the other factor was associated with the cell modification factors, EXPANSIN8 (*EXPA8*) and *EXPA20*, which belong to the expansin gene family and encode enzymes to catalyze the activation of expansin. It is speculated that oxygen plasma might cause acidification of cell wall and might regulate the auxin content [196]. Hayashi et al. conducted a microarray analysis of Arabidopsis seeds and leaves after oxygen and air plasma irradiation. Based on the analysis of GO terms, significantly upregulated genes were shown to belong to antioxidant activity, photosynthesis, and molecular chaperon activation categories. In contrast, genes related to methylation of DNA and DNA replication were downregulated [194]. It was predicted that the plasma irradiation in seeds led to the enhancement of seedling growth due to active oxygen species, suggesting that altered transcriptional regulation is more likely related to epigenetic regulation [38,194].

Recent studies on Arabidopsis utilized RNA-sequencing analysis technology to reveal the transcriptome profile under plasma treatment [194]. Cui et al. performed comparative studies and showed that direct APCP plasma treatment, which generates various types of ROS and RNS, has a dual function of activation and inhibition of seedling growth depending on the treatment duration. Plasma treatment for 1 min in seedlings increased the accumulation of gibberellin and cytokinin, while greatly decreasing the amino acid content. A total of 218 differentially expressed genes (DEGs) were upregulated, whereas 198 genes were downregulated in the RNA-seq analysis. Most of the DEGs were found to be related to defense, stimuli, or stress. Relevant pathways were enriched in MAPK signal transduction, glutathione (GSH) metabolism, and defense related signaling pathways [195]. Waskow et al. found a gene ontology group related to plant stress and defense pathway that is involved in the signaling cascade and impacts the primary and secondary metabolites. The expression of genes related to secondary metabolites were upregulated in 6-day old Arabidopsis seedlings as a long-term memory effect after a brief (60 or 80 s) DBD plasma treatment, which appeared to alter the redox state and induce a response based on overall gene expression pattern [171]. Based on the transcriptome profiling of Arabidopsis, plasma with various reactive species (mainly RONS) seems to play a role as a potential elicitor to regulate stress, plant secondary metabolites, and defense mechanism, although the perturbated genes from RNA sequences are not complete mimics of the genes that are induced under abiotic or biotic stress.

In species other than Arabidopsis, plasma enhances seed germination and growth. The molecular basis for this phenomenon was explained by the increase in the activity of antioxidant enzymes, such as SOD2 and CAT. Additionally, gene expression of adenosine triphosphate (*ATP*), target of rapamycin (*TOR*), and growth-regulating factors (*GRFs*) was upregulated after 15 s of 16.8 kV treatment, although Chinese and American sunflower seeds have different sensitivities to plasma response. In transcriptome analysis of sunflower seeds, 3921 DEGs were found to be involved in starch and sucrose metabolism, DNA replication, and plant hormone signal transduction [170]. Combined treatment with plasma and PAW at 25 kV for 3 h resulted in a survival rate of only 9.2% in seedlings due to damage from severe oxidative stress as the ROS content was high in the seedlings. However, the survived seedlings showed stronger resistance to stress. The upregulated genes from the detected DEGs were associated with transcription, translation, enzyme activity, and metabolism. Moreover, the gene ontology analysis and gene expression profiles indicated that several positive traits, such as redox reaction, chitinase activity, cell proliferation, growth, carotenoid biosynthesis, and growth promoting genes, were upregulated [168].

Intriguingly, plasma irradiation of sunflower seeds was observed using molecular tools, metagenomic analysis, and proteome analysis to generate a plasma mediated-protein network constituting biosynthesis, energy metabolism, protein metabolism, development, and stress. Metagenomic analysis revealed that the plasma generated reactive species barely affected the bacterial diversity and had an impact on the microbial composition to stimulate the growth of roots and lateral organs [197]. Metagenomics and proteomics are beginning to be utilized as novel tools in plasma studies to understand the molecular mechanisms and because zinc oxide (nZnO), selenium (nSe), and silicon (nSi) nanoparticles are gaining attention as new emerging tools for seed priming in the field of plant science [150,152].

### 4.5. Epigenetic and Protein Expression

Since the 2010s, the effect of plasma treatment on DNA methylation has been studied in the biomedical field. Since then, several reports that plasma treatment affects the cells epigenetically and alters the chromatin structure have been published. Short-term effect of plasma treatment occurs due to the epigenetic mechanism involving DNA methylation, in which a methyl group is added to cytosine at the C5 position, forming 5-methylcytosine. DNA methylation modulated gene expression by blocking the binding of transcription factors to DNA or by recruiting proteins that inhibit gene expression [198]. In contrast, bisulfite sequencing analysis showed that plasma treatment affects the increase in the average demethylation level of genes involved in energy production and metabolic pathways related to increase in biomass, which results in higher plant yields [199].

In medical and animal research, it is relatively well-known that plasma treatment triggers DNA methylation and histone modification to modulate gene expression [200,201,202]. In plant species, during seed germination, plasma treatment could affect gene expression, leading to changes in protein expression during germination. Adhikari et al. reported that plasma treatment increases the expression of redox homeostasis genes and alters the epigenetics in tomato [15]. In DNA microarray analysis, seeds irradiated with oxygen plasma showed involvement of photosynthesis and carbon fixation pathways at the transcript level, suggesting that this phenomenon is likely due to epigenetic regulation [38]. In addition to the transcript levels, active oxygen species generated by an oxygen RF plasma modulate the chromatin structure by DNA methylation of HISTONE 1.2 and RNA-directed DNA methylation 4 (*RDM4*) that function as transcriptional regulators and increase the amount of DNA methylation [39].

To explore the overall pattern of alteration in DNA methylation, Perez-Piza et al. used the methylation sensitive amplified polymorphism (MSAP) technique in soybean (*Glycine max*). When treated with PMN3 (Pertinax and Mylar barrier; gas, N_2_; 3 min exposure) and PMO2 (Pertinax and Mylar barrier; gas, O_2_; 2 min exposure), plants showed enhanced growth of shoots and roots. Accelerated phenotypic changes in soybean have been correlated with polymorphic epiallelic changes at the epigenetic level, indicating that DNA methylation pattern variability contributes to superior plant growth [203]. In another study on soybean, the bisulfite sequencing technology was used to investigate the effect of argon plasma treatment at 22.1 kV for 12 s, and an induction of growth was observed. The expression of *ATP a1* (*Glycine max* ATP synthase subunit alpha*)*, *ATP b1*, GRF family members (*GRF 5* and *GRF 6*), and target of rapamycin (*TOR*) was upregulated compared with that in the untreated group, while average methylation levels of these genes were decreased after plasma exposure. The sequenced regions of methylation sites in *ATP a1*, *ATP b1*, *GRF 5*, *GRF 6*, and *TOR* were more frequently found to be of the CG type than the CHG or CHH type after plasma irradiation. Changes in the demethylation levels of the main genes involved in energy metabolism may be responsible for inducing soybean growth [199]. In addition to DNA methylation, histone modification is associated with algal growth and astaxanthin production, which is a blood-red carotenoid with strong antioxidant activity, caused by an increase in plant hormones, ABA and strigolactones, and astaxanthin metabolism related genes. *CRTISO* is a key regulatory gene that affects the synthesis of downstream targets of carotene and astaxanthin. Upregulated *CRTISO* gene expression was the cause of increased histone H3 lysine 4 trimethylation (H3K4me3) in the promoter regions 1 and 2; the presence of regulatory elements at the 5′-end of actively transcribed genes was confirmed using ChiP-PCR. It was confirmed that plasma stimulation enhanced algal growth and astaxanthin accumulation via the upregulation of *CRTISO* gene and increased H3K4me3 in the promoter region of *CRTISO* [204]. H3K4me3 functions as the binding site for recruitment of the SAGA–Chd1 complex and histone acetylation of the SAGA complex regulates transcriptional activation [205]. These results suggest that plasma treatment influences the epigenetic regulation in seeds and seedlings during plant development. However, understanding the molecular mechanisms behind this simultaneous activation and inactivation and coordination in organisms is like an orchestra performance, and the mechanisms by which plasma stimulation triggers epigenetic changes resulting in enhanced germination and growth still needs to be intensively explored with a holistic view of the plasma treatment. Therefore, we have summarized the current knowledge of the molecular mechanisms, including epigenetic regulation, in Figure 3.

### 4.6. Beyond Gene Expression

Plasma offers numerous benefits in increasing seed germination, growth, decontamination, biomass, and secondary metabolites if the plasma source, treatment conditions, and gas feeder are properly applied to the plant. However, it also has harmful negative effects, such as oxidation and genotoxic issues. In barley and pea, prolonged exposure to plasma induced DNA damage, represented by single-strand DNA breaks [206,207,208]. Interestingly, indirect plasma treatment by PAW did not induce DNA damage and improved plant length, fresh weight, and growth parameters in barley and maize [209]. In plasmid DNA, single-strand DNA breaks were observed due to plasma treatment and double strand DNA breaks occurred as the treatment duration increased [210].

Microbial DNA mutagenesis methods were developed in bacteria and fungi [211,212,213]. Although no paper on plant mutagenesis has been published officially, a positive mutation case study was conducted in tickseed flower (*Coreopsis*
*tinctoria* Nutt.) and showed a successful inheritance in the next generation after plasma treatment [214]. This possibility needs to be addressed to utilize the applications of optimal plasma devices for the molecular breeding of plants.

## 5. Conclusions and Future Prospect

In this review, we have summarized the mechanism underlying the effect of plasma on seed germination and plant growth during plant development. In the last few years, the mechanism of plasma treatment of seeds has been revealed based on the evidence of seed surface modification, such as surface scarification and seed coat degradation that leads to increased water uptake by the seeds and accelerates imbibition process. Thus, the physical modifications due to the plasma seed treatment were considered the primary mechanism of plasma treatment, especially in the case of direct treatment. In addition, the studies on plasma treated seed germination were more focused on the chemical interactions between the RONS produced by plasma and plant samples (seeds or seedlings). Moreover, seed sterilization has a positive effect on plant pathogen control. Nevertheless, other physical factors, such as heat, UV irradiation, and magnetic field, cannot be excluded from the parameters explaining the induction of seed germination by plasma treatment. Although the exact molecular mechanisms behind the effect of a single component of ROS or RNS, or the combined effect of both ROS and RNS on seeds and plants remain unclear, the benefits of plasma treatment are established in plant science.

Factors, such as device type and proper configuration of plasma treatment suitable for the particular plant species, variety and ecotype, greatly affect the outcome of the plasma treatment. Investigation of suitable plasma device type is one of the main issues because different machines, power outputs, gas feeders, duration, and other parameters can result in different outcomes. According to the device type, the DBD device is mostly used for the direct plasma treatment of seeds, followed by plasma jets, which are used for both direct and indirect treatments; however, other types of plasma devices are still poorly explored. In addition, direct and indirect treatments have more specific benefits in certain developmental stages of plants. For example, direct treatment is generally more suitable for seed germination, for increasing the germination efficiency and for sterilization of microorganisms, whereas indirect treatment is more effective in enhancing the plant growth, biomass, and yield by PAW irrigation. Therefore, determination of the proper device type for the purpose of the experiment or demand would play an important role in the future. Novel generators developed by physicists are being manufactured according to the demands of biologists, implying that the communication between the experts in physics and biology is important. Moreover, determining optimal conditions through fine-tuning each sample, depending on the plasma treatment duration, concentration and intensity of the active species emitted from plasma, is critical.

At the molecular level, gene expression analysis of specific target genes was first explored by the observation of the effect of plasma treatment. It began with marker genes, such as SOD and CAT, which are known to be involved in the ROS pathway after plasma treatment. Recently, because it is necessary to observe the effect of plasma to have a holistic perspective of the entire genome, an attempt to graft transcriptome and epigenetic profiling has been implemented to understand germination and growth. Future studies should focus on the knowledge gap at the genetic and cellular levels and on the plasma-mediated gene regulatory networks. For example, recently, a few studies used RNA sequencing analysis, which provides a broader molecular view of the regulation and mechanism of plasma treatment at the transcript level. Integration of small RNA analysis, open chromatin profiling, epigenome analysis, proteome analysis, and other techniques will help in interpreting the complex mechanism of plasma treatment.

Furthermore, the characterization of a single gene modulated by plasma, including loss-of-function mutant analysis, will be helpful for in-depth understanding of molecular function in seed germination and plant growth. These trial attempts show the versatile contribution of plasma to the understanding of the basic mechanisms that could be applied in molecular breeding.

LTP has potential as an outstanding novel priming agent for increasing germination, stress resistance, and for promoting useful agronomical traits as well as elicitors by enhancing secondary metabolite production in plants. It represents an eco-friendly and economical disinfectant to sterilize the seed surface that helps in the inhibition of germination retardation and prevents various diseases caused by other pathogens. The potential application of plasma treatment in plants could be as an eco-friendly mutagen instead of a chemical mutagen (such as ethyl methane sulfonate). However, exploring other potential applications of plasma treatment in agriculture is promising and indicates a bright future with the use of plasma technology as a priming agent, elicitor, and infectant to eventually increase stress tolerance, yield and biomass, thus advancing the field of agronomy.

## Figures and Tables

**Figure 1 ijms-23-04609-f001:**
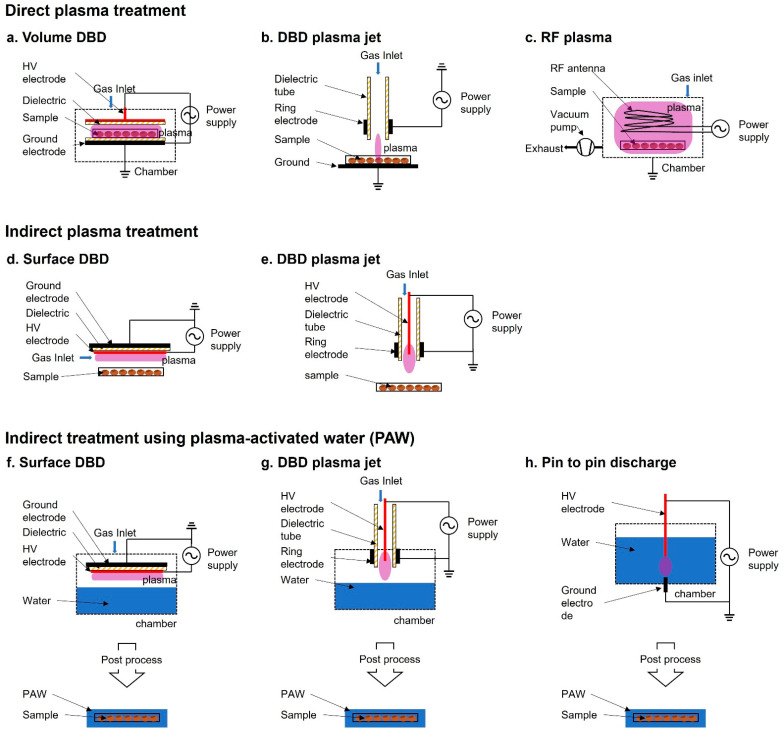
Schematic diagram of plasma devices used for seed treatment. (**a**–**c**) Direct plasma treatment: (**a**) volume dielectric barrier discharge (DBD), (**b**) DBD plasma jet, (**c**) Radiofrequency (RF) plasma; (**d**,**e**) Indirect plasma treatment where the sample is not directly in contact with the plasma discharge: (**d**) surface DBD, (**e**) DBD plasma jet; (**f**–**h**) Indirect treatment using plasma-activated water (PAW), (**f**) surface DBD, (**g**) DBD plasma jet, (**h**) pin-to-pin discharge.

**Figure 2 ijms-23-04609-f002:**
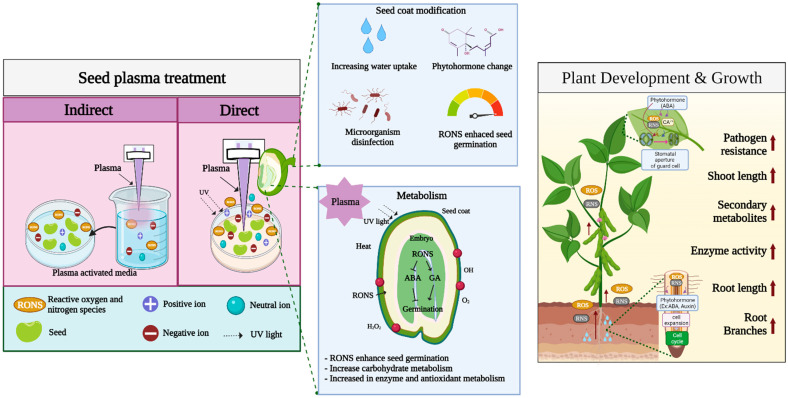
Overview of plasma treatment effects on seed germination and plant growth.

**Figure 3 ijms-23-04609-f003:**
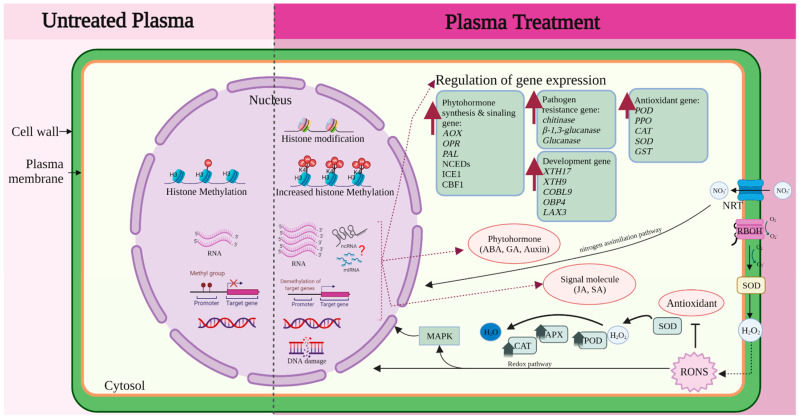
Simplified overview of the molecular mechanisms underlying the effects of plasma treatment. Molecular mechanism underlying the effects of plasma treatment involve complex mechanisms in plant cells including transport activity, signaling molecules, enzyme activity, hormone regulations, gene expression, and DNA and histone modification. Various genes are regulated by redox and nitrogen assimilation pathways.

**Table 1 ijms-23-04609-t001:** Studies on plasma treatment in seeds/plants of different plants species using various plasma devices.

Plasma Source	Gas Feeder	Seeds/Plants	Focus of the Study	Molecular and Physiological	Citation
CCP RF	Air	*Echinacea purpurea* (Coneflower)	Growth,	Vitamin C, Chlorogenic acid, Rutin	[145]
DBD	Argon (Ar)	*Cannabis sativa* (Hemp)	Growth, metabolism	*WRKY1* transcription factor, Cannabinoid genes (*CBDAS*, *THCAS*, *OAC*, *OLS*)	[146]
DBD	Air	*Vigna mungo* L. (Black gram)	Germination and growth	Soluble protein, SOD, CAT	[47]
DBD	Helium (He) and O_2_ (Oxygen)	*Vitis vinifera* cv. Muscat of Alexandria	Seed dormancy	Proline, malondialdehyde (MDA), Catalase	[147]
CCP RF	Air	Maize Wheat Lupine	Growth, metabolism, disease resistance	Phenol, anthocyanin, proline,	[148]
CCP RF	Air	*Helianthus annus* L. (Sunflower)	Growth, metabolism	Phytohormone, protein expression (proteoforms), Protein interaction network	[42]
DBD	Air	*Arabidopsis thaliana*	Growth, Physio-biochemical	NO, nitrate, peroxynitrite, H_2_O_2_ Superoxide content in the seeds. MDA, proline, CAT, SOD, POD. Ca^2+^ expression, atomic ratio	[149]
DBD	Air	Wheat	Growth, metabolism	MDA, NO_3_^−^ content in the seeds	[87]
DBD	Ar	*Astragalus fridae*	Growth, metabolism, nano particle effects, tissue differentiation	Chlorophyll, carotenoid, nitrate reductase, catalase	[150]
DBD	N_2_, O_2_	Soybean	Growth, Biophysical, metabolism	Chlorophyll, MDA, CAT, GPOX, SOD	[151]
DBD	Ar	*Melissa officinalis*	Growth, metabolism, nano particle effects	Peroxidase, PAL	[152]
DBD	Air	Arabidopsis thaliana Col-0, *gl2*, and *gpat5*	Biophysical, growth	*GL2* and *GPAT5* for the embryogenesis and *Peroxidase 69* (AT5G64100)	[37]
RF	O and Ar	*Ocimum basilicum* L. (Basil)	Growth, metabolism	Carbohydrate and protein content, Catalase activity	[153]
Plasma enhanced chemical vapor deposition	Ar	Wheat	Biophysical, growth, disease resistance	Protein content	[154]
DBD	Air, Ar, O_2_	Rapeseed, Mustard	Growth, metabolism	Soluble protein, Chlorophyll, SOD, APX, CAT. Gene expression of BnSOD, BnAPX, BnCAT in root	[48]
DBD	Air, Ar, O_2_	Wheat	Biophysical, growth, metabolism	Chlorophyll content, cadmium, electrolyte, total protein, H_2_O_2_, NO, APX, SOD, CAT. Gene expression of *TaLCT1*, *TaHMA2*, *TaSOD*, *TaAPX*, *TaCAT*	[155]
DBD	Air, Ar, O_2_	Wheat	Biophysical	-	[156]
RF	N_2_, O_2_	Asparagus	Biophysical, growth	-	[157]
ICCP RF	He	Tomato	Growth	N, P, K content	[158]
CCP RF	N_2_	Artichoke	Biophysical, growth, metabolism	POD, CAT content	[159]
DBD	He	Coffee, grape seeds	Growth	-	[49]
Gliding arc	Air	Pea Zucchini	Growth	-	[160]
DBD	Air	Thurigan Mallow	Biophysical, growth	-	[50]
LFGD	Ar and Air	maize (*Zea mays* L.)	Biophysical, growth	CAT, SOD, APX, GR content, total soluble protein, sugar, fat content, chlorophyll	[161]
DBD	Air	Rice		Gene expression of ABA catabolism (*OsABA8′OH1−3*), ABA biosynthesis genes (*OsNCED1−5*), α-amylase gene (*OsAmy1–3*),	[162]
DBD	N_2_, O_2_	Soybean	Biophysical, metabolism	Phytohormones (indole acetic acid/IAA and trans-zeatin riboside/tZR) glutathione, nitrogen activity, leghemoglobin content in nodule and *GmEXP1* gene expression in roots	[163]
DBD	Air	Bitter melon (*Momordica charantia*)	Growth	-	[164]
DBD	Air	Tomato	Chilling resistance, metabolism	Phytohormone, hydrogen peroxide and abscisic acid signaling gene (*RBOH1*, *NCED1*, *NCED2*, *ICE1*, *CBF1*)	[165]
DBD	Ar	*Astragalus fridae*	Biophysical, metabolism	Chlorophyll, expression of phenylalanine ammonia lyase (*PAL*), and universal stress protein (*USP*).	[150]
DBD	Ar	*Catharanthus roseus*	Biophysical, metabolism	Chlorophyll, carotenoids, phenylalanine ammonia lyase (PAL) enzyme and deacetyl vindoline O-acetyltransferase (*DAT*) gene	[166]
DBD	Air	*Andrographis paniculata*	Biophysical, growth	Expression gene related to plants hormone (*ACO*, *NCED5*, *CRF4*, *NRT1*, *RAP2-10*, *ERF098*, and *PRP**3*)	[167]
Needle array plate DBD	Air	*Astragalus adsurgens* Pall.	Biophysical, metabolism	-	[168]
Plasma Air-jet	Air	Tomato	Biophysical, metabolism	Antioxidants, phytohormones, and expression of defense genes (*POD*, *CAT*, *PPO*, *SOD*, *GST*, *AOX*, *OPR*, *PAL*, *HAT*, *HFMET*, *MPK*, and *RBOH*)	[15]
DBD	Air	*Panax ginseng*	Biophysical, metabolism, pathogenic defense system	Chlorophyll, total phenolic and pathogen-associated gene (*PgPR2*, *PgPR5*, *PgPR10*, and *PgCAT*)	[169]
DBD	Ar	*Helianthus annus* L. (Sunflower)	Biophysical, metabolism	Protein content, antioxidant enzyme activity, and RNA-seq	[170]
SDBD	Air	*Arabidopsis thaliana*	Plant defense	RNA-seq young seedling	[171]
SDBD	Air	Tobacco (*Nicotiana tabacum*)	Plant development, gene expression	Morphology, root and root hair development genes (*NtCOBL*, *NtXTH5*, *NtXTH9*, *NtXTH15*, *NtXTH17*, *NtXTH27*)	[73]
SDBD	Air	Arabidopsis	Plant development, gene expression	Morphology, root and root hair development genes (*AtCOBL9*, *AtAUX1*, *AtLAX3*, *AtOBP4*, *AtXTH17*)	[72]
DBD	Air	Wheat	Germination, growth, metabolism	CAT, SOD, gene expression (*TaCAT*, *TaSOD*)	[88]
DBD	Ar	Fenugreek (*Trigonella foenum-graecum*)	Metabolism, secondary metabolite, gene expression	Content of protein, carotenoid, chlorophyll, CAT, GPX, APX. Gene expression (*SEP*, SQS, *CAS*, *SSR*, *SMT*)	[172]
DBD	Air	Tomato	Plant defense, gene expression	Expression of defense genes (*PAL*, *ERF1*, *PR1a*, *PR4*, *PR5*)	[173]
Diffuse Coplanar Surface Barrier Discharge (DCSBD)	Air, O_2_, N_2_	Maize	Plant growth, metabolism, gene expression	Enzyme activity (protease, glucanase, POX, SOD), expression of heat shock protein genes (*HSP101*, *HSP70*, *HSF17*)	[137]

## Data Availability

Not applicable.

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
