# Peer review of "Current Advancements in the Molecular Mechanism of Plasma Treatment for Seed Germination and Plant Growth"

_ijms, 2022, doi:10.3390/ijms23094609_

Round 1

Reviewer 1 Report

The Authors have treated an intersting topic for the improvement of seed germination and plant growth.

Their review is complete and well organized.

The have treated extensively the technical aspects and also the biological implication.

The Authors have worked extensively to resume the actual knowledge about plasma treted.

Their English language is good

In my opinion this paper will be useful for the scientist which work in this field

I recommend the pubblication

Author Response

Thank you for your favorable comments. We appreciate you for the time expended in reviewing our manuscript. We hope that this article would contribute to the scientific community, especially in the field of plasma application and molecular studies.  

Reviewer 2 Report

Review on Current Advancement in the Molecular Mechanism of Plasma Treatment for Seed Germination and Plant Growth

I have completed my review on manuscript ijms-1683832, entitled, Current Advancement in the Molecular Mechanism of Plasma Treatment for Seed Germination and Plant Growth It is proved in the last few decades that plasma is a potential candidate to boost the plant growth and seed germination in various ways. The literature reviewed by the authors in this manuscript is useful for research in plasma agriculture.

Overall, it was very informative review article about plasma and its future perspective in the agriculture field. I have some comments on the present form of this manuscript, the manuscript deserves to publish after the recommended major revisions.

Comment 1. In Figure 1 (b),

(A).      For DBD discharge it is necessary that the power and ground electrodes are separated by a dielectric material. Figure 1 (b) shows now the dielectric barrier between power and ground electrode, in that case, maybe it's not DBD discharge. The figure needs to modify if it is a mistake or if the figure is correct authors need to explain how it can make DBD discharge when power and ground electrodes' faces can see each other without a dielectric barrier.

(B).      Also, the gas inlet needs to be mentioned in all schematics as this information is missing from Figs 1 (a), (d), and (f).

(C).      In plasma sources, if samples are placed above the ground, it is a direct plasma treatment source as in Figure 1 (a), and if the sample is placed after the ground, it is an indirect plasma source. I think the authors need to correct this as it is not correct to say “indirect treatment using plasma-treated gas”. What does it mean by “plasma-treated gas”?

Comment 2. I recommend authors add literature based on new techniques (between the last 2 years). Some plants are sensitive to pH and plasma decreases the pH of water normally. Recently, an idea was suggested to control the pH of water after plasma (https://doi.org/10.3390/ijms22105360) which proved to be very useful in agriculture. I recommend authors add some literature by describing the relation between plasma and pH.

Comment 3: In section 3.1 line no 230, it was mentioned the generation of ozone in plasma. Ozone up to a certain percentage plays a beneficial role after that it will gives negative effects on plant growth. Can you please summarize some literature about what percentage is useful for plants?

Comment 4: In section 3.1, the authors mentioned the role of ROS and RNS in plant growth. I recommend authors mention the appropriate concentrations because an excessive amount of ROS and RNS might possibly have bad effect. If possible, please mention the suitable concentration of ROS and RNS which are commonly known to have useful effects on plant growth.

Comment 5: In line 254, the authors mentioned microbes’ inactivation by plasma. In which you mentioned fungal spores also. Can you review the possible idea for virus inactivation by plasma? This is very necessary for the current situation possible with some mechanisms from literature.

Comment 6: In figure 2 mentioned direct and indirect plasma treatment. Can you please justify which treatment is more beneficial and effective for plant growth?

Comment 7: Section 4.3 line 533, the authors mentioned about MAPK pathway. MAPK pathway consists of three family members: P38, JNK, and ERK. Each has a different role in different conditions. Can you please explain some detail about how it will helpful for plant germination through the MAPK family members?

Comment 8: There are typos and inaccuracies in the paper, I recommend authors to read precisely and correct the grammatical errors.

Author Response

I have completed my review on manuscript ijms-1683832, entitled, “Current Advancement in the Molecular Mechanism of Plasma Treatment for Seed Germination and Plant Growth” It is proved in the last few decades that plasma is a potential candidate to boost the plant growth and seed germination in various ways. The literature reviewed by the authors in this manuscript is useful for research in plasma agriculture.

 Overall, it was very informative review article about plasma and its future perspective in the agriculture field. I have some comments on the present form of this manuscript, the manuscript deserves to publish after the recommended major revisions.

> We appreciate the time and effort put by the reviewer in thoroughly reviewing our manuscript. Please find our responses to each of your comments below. The responses are in bold blue font and the revised text is highlighted in yellow in the manuscript.

Comment 1. In Figure 1 (b),

(A).      For DBD discharge it is necessary that the power and ground electrodes are separated by a dielectric material. Figure 1 (b) shows now the dielectric barrier between power and ground electrode, in that case, maybe it's not DBD discharge. The figure needs to modify if it is a mistake or if the figure is correct authors need to explain how it can make DBD discharge when power and ground electrodes' faces can see each other without a dielectric barrier.

> Thank you for pointing out this mistake. We have made the necessary corrections in Figure 1(b).

(B).      Also, the gas inlet needs to be mentioned in all schematics as this information is missing from Figs 1 (a), (d), and (f).

> As suggested, we have mentioned the gas inlets in Figure 1 (a-h).

(C).      In plasma sources, if samples are placed above the ground, it is a direct plasma treatment source as in Figure 1 (a), and if the sample is placed after the ground, it is an indirect plasma source. I think the authors need to correct this as it is not correct to say “indirect treatment using plasma-treated gas”. What does it mean by “plasma-treated gas”?

> For classifying indirect plasma treatment, ”plasma-treated gas” and “plasma-activated water” were written to distinguish the plasma treatment using gas-phase active species and liquid-phase active species. In view of your comment, we have replaced “Indirect treatment using plasma-treated gas” with “Indirect plasma treatment” for clarity.

Comment 2. I recommend authors add literature based on new techniques (between the last 2 years). Some plants are sensitive to pH and plasma decreases the pH of water normally. Recently, an idea was suggested to control the pH of water after plasma (https://doi.org/10.3390/ijms22105360) which proved to be very useful in agriculture. I recommend authors add some literature by describing the relation between plasma and pH.

> Thank you for the information and for your valuable recommendation. We agree that a low pH is harmful to plants and poses one of the main challenges in the plasma-treated method, especially using plasma-activated water. We had cited the article on plasma-assisted nitrogen fixation for corn in the original manuscript (line 272). We have included additional description in the revised manuscript, as mentioned below (lines 277–287).

 “However, one of the challenges in plasma-assisted nitrogen fixation is the low pH or increased acidity of the solution treated with plasma, which damages the seed and plant exposed to such solutions. Plant growth is limited under acidic environment [94]. Therefore, studies on controlling the balance and on methods to overcome the acidity of plasma-activated solutions are being considered on priority in the plasma field. Lamichhane et al. recently demonstrated an innovative approach to control the acidity of plasma-treated water using a combination of chemical additives including Mg, Al, or Zn, which could neutralize the reduction in pH. Moreover, the presence of these additives increases the rate of reduction of nitrogen to ammonia, which results in the improvement of germination rate and seedling growth [93].”

Comment 3: In section 3.1 line no 230, it was mentioned the generation of ozone in plasma. Ozone up to a certain percentage plays a beneficial role after that it will gives negative effects on plant growth. Can you please summarize some literature about what percentage is useful for plants?

> Thank you for the pertinent question. It is correct that depending on its amount ozone has different effects on plasma-treated samples. As discussed in the manuscript, at the proper amount ozone has beneficial effects on the samples, whereas excessive amounts increase the potential of damage. We have further discussed this matter and cited several new articles. The revised text reads as follows (lines 247–260):

 “However, it is important to note that different device configurations produce different concentrations of ozone. Moreover, the purpose of treatment also determines the ozone concentration required. Postharvest treatment with 0.3 ppm ozone in combination with cold storage could inhibit the decay process and reduced severe infection in peach and table grapes [81,82]. In strawberry, plasma treatment using different sources of gas has been investigated; it was observed that plasma treatment for 5 min. could produce 600–2800 ppm ozone that had a positive effect on microbial disinfection and strawberry freshness [83]. Similarly, various ozone concentrations were investigated in the plasma treatment of seeds. In Arabidopsis seeds, the effects of treatment with 200 ppm ozone, generated from a plasma device, for 10 min on seed coat modification were studied [84]. Low concentration of ozone (~1–5 ppm) is effective in plant growth by killing larvae in the soil and in fresh cut green leaf lettuce [85]. Interestingly, a similar concentration of ozone (~1–4 ppm) generated from surface discharge successfully reduced the number of nematodes and induced plant growth [86].”

Comment 4: In section 3.1, the authors mentioned the role of ROS and RNS in plant growth. I recommend authors mention the appropriate concentrations because an excessive amount of ROS and RNS might possibly have bad effect. If possible, please mention the suitable concentration of ROS and RNS which are commonly known to have useful effects on plant growth.

> Thank you for the suggestion. In the revised manuscript, we have added more details regarding the ROS and RNS concentrations. The following text has been added to the manuscript (lines 221–227):

“A relatively small amount of H2O2 (0.12 ppm) in PAW, generated from a plasma device, increased the germination rate in tomato and pepper seeds [20]. In Arabidopsis, PAW containing 17–25.5 mg/L H2O2 had a positive effect on germination and seedling growth [14]. Among RNS, low nitrate concentration (100 ppm or less) enhances seed germination and seedling growth in plants, but the growth tends to be inhibited above 100 ppm [20,72–74]. Moreover, plants apparently have their own nitrate and ROS sensitivity and show a dosage-dependent growth pattern.”

Comment 5: In line 254, the authors mentioned microbes’ inactivation by plasma. In which you mentioned fungal spores also. Can you review the possible idea for virus inactivation by plasma? This is very necessary for the current situation possible with some mechanisms from literature.

> Thank you for the suggestion. We agree that recent plasma treatment for virus inactivation has become one of the hot topics in this field. We have added information about virus inactivation by plasma treatment in the revised manuscript. Furthermore, we have also mentioned about the application of plasma treatment in inactivation of plant viruses referring to several articles and have described the possible mechanism. Please refer to the following text in the revised manuscript (lines 328–338):

“Other applications of plasma treatment include inactivation of viruses. In the field of plasma biomedicine, the inactivation of coronavirus using plasma devices has recently garnered a lot of attention and proved to be effective in treating viral infection and associated diseases [113–115]. In plants, potential applications of plasma treatment of virus-related diseases have been reported. Potato virus Y (PVY) homogenized in water was successfully inactivated by plasma treatment for 1 min. [116]. A sample of pepper mild mottle virus (PMMoV) in water was also successfully inactivated using 99% argon and 1% oxygen plasma jet in 5 and 3 min. [117]. As for the inactivation of bacteria, the mechanism of virus inactivation is primarily through the production of reactive species (ROS and RNS) with various physical effects, and the treatment could damage the virus particle and degrade viral DNA/RNA [118,119].”

Comment 6: In figure 2 mentioned direct and indirect plasma treatment. Can you please justify which treatment is more beneficial and effective for plant growth? 

> Thank you for the question. As mentioned in section 5, a proper device type and configuration is important in determining the final goal of the plasma treatment. Therefore, direct treatment or indirect treatment can be utilized depending on the purpose. To our knowledge, direct treatment is generally more beneficial in the seed germinations steps because of seed surface modification and inactivation of microbes. Indirect treatment has been majorly reported as the generation of plasma-activated water that contains a high content of nitrogen. In addition, indirect treatment through water irrigation not only increases the growth of plants but also induces their defense capabilities. Therefore, we believe that indirect treatment with plasma-activated water irrigation is more effective for plant growth.  We have added the following discussion in section 5 (Lines 805–814):

“According to the device type, DBD device is mostly used for direct plasma treatment of seeds, followed by plasma jets, which are used for both direct and indirect treatments; however, other types of plasma devices are still poorly explored. In addition, direct and indirect treatments have more specific benefits in certain developmental stages of plants. For example, direct treatment is generally more suitable for seed germination, for increasing the germination efficiency, and for sterilization of microorganisms, whereas indirect treatment is more effective in enhancing the plant growth, biomass, and yield by PAW irrigation. Therefore, determination of the proper device type for the purpose of the experiment or demand would play an important role in the future.”

Comment 7: Section 4.3 line 533, the authors mentioned about MAPK pathway. MAPK pathway consists of three family members: P38, JNK, and ERK. Each has a different role in different conditions. Can you please explain some detail about how it will be helpful for plant germination through the MAPK family members?

> Thank you for the question. To our understanding, the detailed mechanism for the involvement of the MAPK pathway in plasma treatment has been well characterized in the biomedical field, especially for animals and for the treatment of cancer. Furthermore, the MAPK pathway members that you have mentioned are generally categorized in the mammalian cells. We have described about the role of MAPKs in plants in the manuscript (lines 586–594) and have added some details in the revised Figure 3. 

“Plant MAPKs are more similar to the mammalian extracellular signal-regulated kinase (ERK) subfamily of MAPKs and their cascade plays a vital role in plant defense signaling under biotic and abiotic stress. MAPKs include the signaling molecules—MAPK kinase kinase (MAPKKKs) activate MAPK kinase (MAPKKs), which eventually activate MAPKs [189,190]. H2O2 transduces signal via calcium as the secondary messenger and the MAPK cascade. RONS (NO and H2O2) produced by PAW irrigation induce plant hormones, SA and JA, and PR proteins via the MAPK signaling. At transcriptional level, upregulation of MAPK related gene expression was observed under PAW irrigation [68,180].”

Comment 8: There are typos and inaccuracies in the paper, I recommend authors to read precisely and correct the grammatical errors.

 > We apologize for the inaccuracies and typographical errors. We have thoroughly checked the revised manuscript for accuracy. In addition, this revised version has been edited by a professional English editing service.

Round 2

Reviewer 2 Report

Review on the revised version of, "Current advancements in the molecular mechanism of plasma treatment for seed germination and plant growth."

I appreciate the way the authors addressed my comments and concern. The revised version of the manuscript deserves to be published in IJMS in the present form.